# Working memory performance is tied to stimulus complexity

Roland Pusch [1,8 ✉], Julian Packheiser[1,2,8], Amir Hossein Azizi[3], Celil Semih Sevincik[1], Jonas Rose[4], Sen Cheng[5], Maik C. Stüttgen [6] & Onur Güntürkün[1,7]

Working memory is the cognitive capability to maintain and process information over short periods. Behavioral and computational studies have shown that visual information is associated with working memory performance. However, the underlying neural correlates remain unknown. To identify how visual information affects working memory performance, we conducted behavioral experiments in pigeons (*Columba livia*) and single unit recordings in the avian prefrontal analog, the nidopallium caudolaterale (NCL). Complex pictures featuring luminance, spatial and color information, were associated with higher working memory performance compared to uniform gray pictures in conjunction with distinct neural coding patterns. For complex pictures, we found a multiplexed neuronal code displaying visual and value-related features that switched to a representation of the upcoming choice during a delay period. When processing gray stimuli, NCL neurons did not multiplex and exclusively represented the choice already during stimulus presentation and throughout the delay period. The prolonged representation possibly resulted in a decay of the memory trace ultimately leading to a decrease in performance. In conclusion, we found that high stimulus complexity is associated with neuronal multiplexing of the working memory representation possibly allowing a facilitated read-out of the neural code resulting in enhancement of working memory performance.

[1] Department of Biopsychology, Faculty of Psychology, Ruhr University Bochum, Universitätsstraße 150, D-44780 Bochum, Germany. [2] Social Brain Lab, Netherlands Institute for Neuroscience, Amsterdam, The Netherlands. [3] Department of Systems Biology, Agricultural Biotechnology Research Institute of Iran (ABRII), Karaj, Iran. [4] Neural Basis of Learning, Faculty of Psychology, Ruhr University Bochum, Universitätsstraße 150, D-44780 Bochum, Germany. [5] Institute for Neural Computation, Faculty of Computer Science, Ruhr University Bochum, Universitätsstraße 150, D-44780 Bochum, Germany. [6] Institute of Pathophysiology, University Medical Center of the Johannes Gutenberg University, Duesbergweg 6, D-55128 Mainz, Germany. [7] Research Center One Health Ruhr, Research Alliance Ruhr, Ruhr University Bochum, Bochum, Germany. [8] These authors contributed equally: Roland Pusch, Julian Packheiser.
✉email: roland.pusch@rub.de

Working memory, the ability to maintain and process information in the absence of sensory stimuli, is crucial for many cognitive functions like abstract reasoning, problem solving, goal-directed behavior, decision making, and cognitive control[1]. This important role in cognition is further illustrated by the fact that the capacity of working memory is often taken as proxy for overall cognitive capacity[2,3]. A large body of research has demonstrated that especially the prefrontal cortex (PFC) is involved in the active maintenance of sensory input[4–6], for reviews see refs. [7,8]. Most studies investigating the neural basis of working memory focused on the specific effects of increased working memory load[9,10], for review see ref. [11], influence of reward associations[12,13] or the role of attention[14,15]. Surprisingly, the stimulus that is used in a working memory paradigm has received comparatively little attention in neuroscience.

Across a series of behavioral studies in humans, preliminary evidence has been collected that the stimulus material impacts working memory performance. In a behavioral study, Gegenfurtner and Rieger[16] investigated the impact of color on encoding and retrieval processes. They found that recognition and recall of colored stimuli was faster and more successful than recognition of black and white images. Bae et al.[17,18] investigated the precision of working memory in delayed estimation tasks using plain colors as stimuli. In these experiments, a target color had to be remembered throughout a retention interval. Subsequently, the target color had to be recalled and estimated on a continuous color wheel. The deviation of the chosen color from the presented stimulus indicated the memory precision for a specific hue. Importantly, the authors found an inhomogeneous response distribution where specific colors were maintained more precisely than others, even when remembering a single stimulus. These results were later replicated in humans and monkeys and modelled as drift towards adaptive attractors. According to these models, neural representations in working memory are more resistant to noise when they are stored in stable states associated with, for example, primary colors[19]. Similar findings were made by Pratte et al.[20] using the orientations of gratings as stimuli. As for color vision, the authors found an inhomogeneous distribution for the recall of orientations with a superior memory performance for cardinal orientations. The abovementioned experimental and modeling work indicates the influence of stimulus material on memory performance, a notion that is further illustrated by the interaction between working memory and perception[21].

While these recent behavioral and modeling studies have demonstrated that the stimulus material indeed affects working memory, the possible underlying neural computations remain elusive. To provide insights into these processes, we investigated the impact of stimulus complexity on working memory performance in three behavioral experiments. As stimuli of low complexity, we used uniform gray pictures that only varied along one dimension: their luminance. Complex stimuli were pictures that varied along several stimulus dimensions. These pictures were composites of elemental features providing luminance, spatial and color information. Varying complexity, defined as the "amount of visual detail" by Alvarez and Cavanagh[22], of the stimulus material has been shown to impact working memory performance in human subjects[22–24]. Here, we chose pigeons as a model organism for the present study due to their excellent color vision[25], their ability to quickly learn and memorize visual stimuli[26], and to categorize pictures on par with monkeys[27]. Further, all investigated functional parameters of working memory have been suggested to be comparable between birds and mammals e.g. refs. [27,28]. Across all behavioral experiments, we consistently found higher working memory performance rates for high

compared to low stimulus complexity. In a final experiment, we complemented these behavioral findings with single-unit recordings from the nidopallium caudolaterale (NCL). This structure has been suggested to be the avian analogue to the mammalian PFC based on neurochemical results[29], connectivity patterns e.g. ref. [30] and its role in executive functions e.g. ref. [31], especially working memory[32–34]. Here, we found a systemic difference in the neural representation of choice timing and dimensionality between the two stimulus classes.

## Results

**Behavioral experiments reveal effects of stimulus complexity on working memory performance.** In three behavioral experiments, we investigated the effect of stimulus complexity on working memory by comparing performance accuracy with and without a delay in a paired association task. All trials in the experiments followed the same structure (see Fig. 1a and Methods *Behavioral paradigm* for details): After a center peck to initialize the trial, a sample phase of 1s duration began, during which a single stimulus was presented on the center response-key. In the non-delayed condition, the choice period started after another peck onto a confirmation key following the sample phase. In the delayed condition, a 3 s delay phase was imposed between the sample and the choice phase. During the choice phase, the animals had up to 3 s to make a choice either to peck on the left or on the right response key (counterbalanced across animals). A correct choice resulted in a 2 s food presentation whereas an incorrect choice resulted in a mild punishment with a 2 s lights out condition in the experimental chamber. In each experiment, eight different images were presented as stimuli. Four 'complex' stimuli consisted of multiple colors as well as spatial features and four 'simple' stimuli consisted of a single grayscale value of different luminance (Fig. 1b, left panel). We will use the terminology complex vs. simple throughout the manuscript to refer to the two stimulus classes (for more detail on stimulus characterization see Methods *Stimuli*).

**Experiment 1:** The first experiment had two aims: First, we wanted to establish if the animals could discriminate equally well between complex and simple stimuli. For this purpose, we analyzed the non-delayed condition of our task. Our second aim was to explore the effects of stimulus-complexity on working memory performance. For this purpose, we compared the results from the non-delayed with the delayed condition of our task. In experiment 1, the animals associated each stimulus with either the left or the right response key. All stimuli were rewarded in 100% of the cases if the correct choice was made (Fig. 1b, Experiment 1).

89 behavioral sessions from 11 animals were analyzed (see Supplementary Table 1 for a detailed description). To identify whether performance was affected by stimulus complexity, we compared the number of correct responses between complex and simple stimuli in the non-delayed condition using a linear mixed model. The model contained "delay" (yes/no as levels) and "class" (levels: complex stimuli, simple stimuli) as fixed effects and the individual animal as a random effect to account for the hierarchical nature of the data. There was a significant interaction effect between the factors "class" and "delay" ($\beta = -0.07$, 95% CI $= [-0.09$ to $-0.04]$, SE $= 0.01$, $p < 0.001$). We found no significant difference between the performance for complex stimuli (93.5% correct responses) and the performance for simple stimuli (92.4% correct responses, $t = 1.37$, $p = 0.173$, $d = 0.27$, 95% CI $= [-0.12$ to $0.67]$) in the "no delay" condition indicating that complex and simple stimuli could be discriminated equally well if no working memory component was present in the task (Fig. 2a). In the "delay" condition, however, we found a much higher performance for complex stimuli (91.4% correct

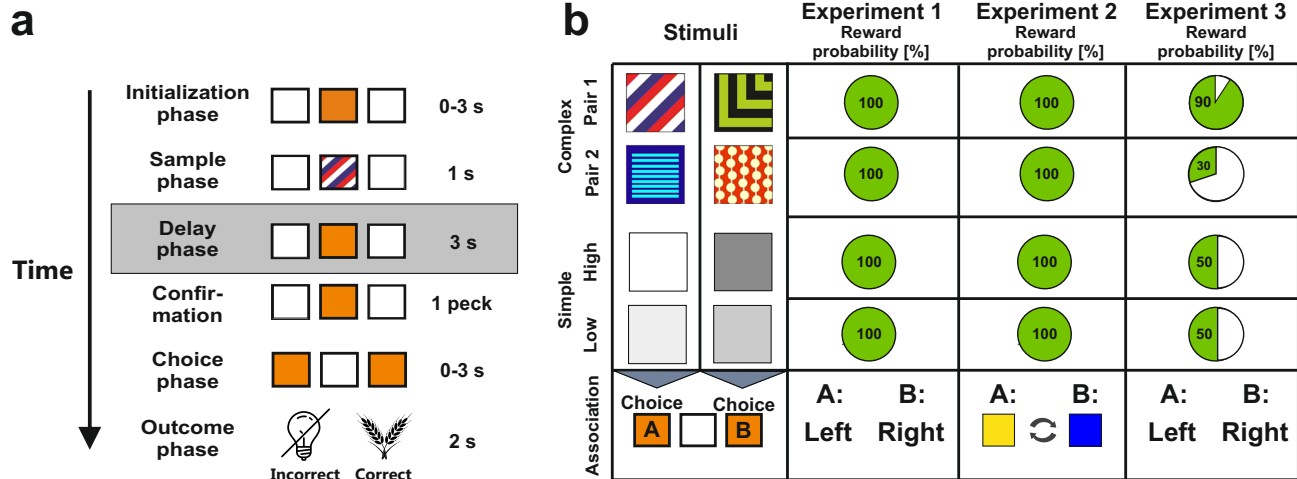

**Fig. 1 Behavioral paradigm, stimuli, and experimental conditions. a** Depiction of the paired association task used during each experiment. The boxes represent the pecking keys in the operant chamber and their illumination for each phase. On the right, the duration of each phase is stated. The gray frame around the Delay phase indicates that it was only present in the delayed condition. **b** In the left panel, the stimuli used in our paradigm are depicted. In the panels to the right, reward probabilities for the stimuli across the experiments are presented. In experiment 1 and 2, all stimuli were followed by food for every correct choice. In experiment 3 and during the electrophysiological recordings, correct responses were rewarded with different probabilities. The bottom row demonstrates what choices had to be made in each experiment. Experiment 1 and 3 required spatially fixed choices associated with each stimulus (choice A = left choice, choice B = right choice). The stimulus-choice associations were counterbalanced across stimuli. In Experiment 2, the stimuli were associated with a specific target color (choice A = yellow, choice B = blue) that varied in its spatial location from trial to trial during the experimental session.

responses) compared to simple stimuli (83.4% correct responses, $t = 8.61$, $p < 0.001$, $d = 1.95$, 95% CI = [1.46–2.44]). Resolving the interaction for the factor "delay" showed a significant reduction for both complex ($t = 2.62$, $p = 0.001$, $d = 0.61$, 95% CI = [0.15–1.08]) and simple stimuli performance ($t = 9.80$, $p < 0.001$, $d = 2.29$, 95% CI = [1.77–2.81]).

Of important note for a later experiment is that there was a difference in performance between simple stimuli that were separated by low and high contrast increments in both experimental conditions, i.e. with and without a delay. Low contrast simple stimuli received fewer correct responses compared to stimuli separated by higher contrast differences (high contrast = 90.6% correct responses; low contrast = 86.4% correct responses, $t = 4.71$, $p < 0.001$, $d = 0.71$, 95% CI = [0.41–1.01]). However, the performance decline for both high and low contrast stimuli was uniform across the delay indicating that this effect was likely working memory independent (high contrast difference: $t = 7.74$, $p < 0.001$, $d = 1.74$, 95% CI = [1.28–2.20]; low contrast differences: $t = 8.20$, $p < 0.001$, $d = 1.84$, 95% CI = [1.38–2.30], see Supplementary Table 2 for descriptive data). Results for the individual stimuli are presented in Supplementary Fig. 1.

**Experiment 2:** In the first experiment, we found that complex stimuli could be maintained more accurately over time compared to simple stimuli. However, the task in experiment 1 demanded a spatially fixed choice to the left or to the right depending on the stimulus identity. To control for spatial memory effects or motor preparation strategies, e.g. preemptively locating oneself closer to the respective choice target, we repeated the experiment, but replaced the spatially fixed response associations with randomly presented color associations (choice A = yellow, choice B = blue, counterbalanced across animals; Fig. 1b, Experiment 2). Thus, the target location for the upcoming choice could no longer be predicted by the animals.

39 behavioral sessions from eight animals were analyzed for experiment 2 (see Supplementary Table 1 for a detailed description). As in experiment 1, we found differential effects of

the delay depending on the stimulus class (interaction of class*delay: $\beta = -0.08$, 95% CI = [−0.12 to −0.04], SE = 0.02, $p < 0.001$, Fig. 2b). We did not find a significant difference in the "no delay" condition between the complex (94.5% correct responses) and simple stimuli (94.2% correct responses, $t = 0.27$, $p = 0.790$, $d = 0.08$, 95% CI = [−0.53 to 0.70]). For the "delay" condition, however, we found a significant performance difference between both stimulus classes (complex: 93.0% correct responses; simple: 84.3% correct responses, $t = 5.84$, $p < 0.001$, $d = 1.95$, 95% CI = [1.21–2.69]). Identically to experiment 1, we again resolved the interaction for the factor "delay". We found no significant reduction in performance for complex ($t = 1.89$, $p = 0.062$, $d = 0.63$, 95% CI = [−0.04 to 1.31]) but a significant reduction for simple stimuli ($t = 7.46$, $p < 0.001$, $d = 2.50$, 95% CI = [1.71–3.82]). These findings strongly resemble the findings from the first experiment and show that complex stimuli were maintained more accurately than simple stimuli. Importantly, this effect could not be explained by spatial memory effects or motor preparation strategies (see Supplementary Table 3 for descriptive data). Results for the individual stimuli are presented in Supplementary Fig. 2.

**Experiment 3:** In experiment 1 and in experiment 2, we found that simple stimuli separated by low contrast differences received fewer correct responses compared to simple stimuli separated by higher contrast differences. This effect could be purely rooted in perceptual differences between the simple stimuli or concomitantly by the resulting reward differences. Overall animals received less food when confronted with low compared to high contrast stimuli resulting in different reward net outcomes. To disentangle perceptual and reward involvement as the driving force of the decline in choice behavior, we modulated the reward contingencies in experiment 3: One pair of complex stimuli was subsequently associated with a high reward probability (90%), while the other pair of complex stimuli was associated with a low probability (30%, counterbalanced across animals). The clear divergence in reward probability was chosen to achieve a pronounced difference in

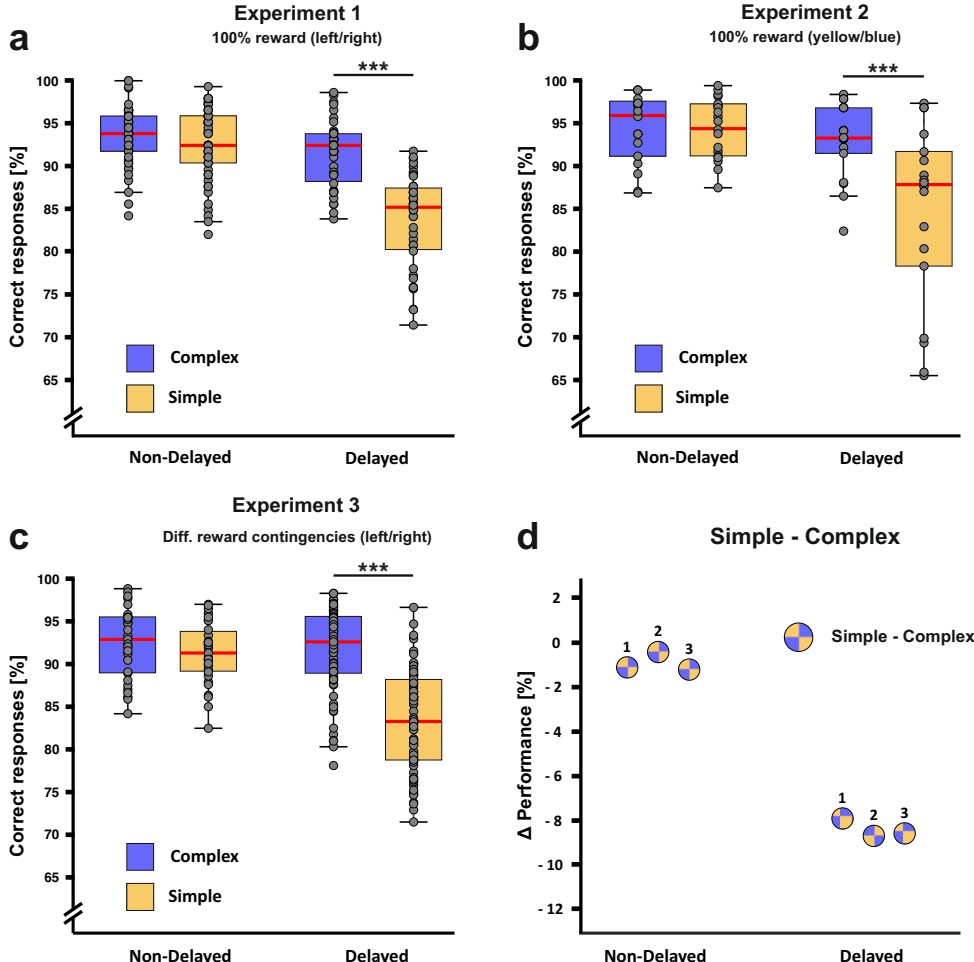

**Fig. 2 Behavioral performance for the experiments 1–3. a** Experiment 1: In the non-delayed condition, no performance difference was found between complex (blue) and simple stimuli (yellow) indicating that both stimulus classes could be successfully associated with their corresponding choice if no working memory component was present. Following the introduction of the delay, a much larger performance drop was observed for simple compared to complex stimuli. Dots represent individual sessions in the respective experimental conditions. **b** Experiment 2: Dissociating the upcoming choice from a fixed spatial location had no effects on the result pattern that was observed in experiment 1. **c** Experiment 3: As in experiment 1 and 2, performance decreases following the introduction of the delay were much more pronounced for simple compared to complex stimuli despite changed reward contingencies for complex stimuli. **d** Percentage difference (Δ) of the performance between the delayed vs. the non-delayed condition for complex and simple stimuli. The numbers next to the circles indicate the performance difference in the specific experiment. ***$p < 0.001$. Error bars represent the 95% confidence interval.

the overall net outcome among the complex stimuli. Reward probabilities for the simple stimuli were kept at identical rates of 50% to enhance the effect of net outcome difference via reward probability and to directly compare the discrimination difficulty (Fig. 1b, Experiment 3).

104 behavioral sessions from 17 animals were analyzed for experiment 3 (see Supplementary Table 1 for a detailed description). Reiterating the effects in experiment 1 and 2, we found a differential effect of the introduction of the delay across the two stimulus classes (interaction of class*delay: $\beta = -0.07$, 95% CI = [−0.10 to −0.04], SE = 0.01, $p < 0.001$, Fig. 2c). Even though we modulated reward probabilities of the complex stimuli, we again found no difference between complex (92.4% correct responses) and simple stimuli (91.2% correct responses) without a delay ($t = 0.97$, $p = 0.335$, $d = 0.25$, 95% CI = [−0.26 to 0.75]). As for the first two experiments, introducing a delay to the experimental conditions revealed a strong performance difference between the two stimulus classes (complex: 91.7% correct responses; simple: 83.2% correct responses, $t = 10.64$, $p < 0.001$, $d = 1.76$, 95% CI = [1.39–2.13]). As in experiment 2, resolving

the interaction for the factor "delay" showed no significant reduction for complex ($t = 0.39$, $p = 0.700$, $d = 0.11$, 95% CI = [−0.67 to 0.46]) but a significant reduction for simple stimuli in performance ($t = 4.99$, $p < 0.001$, $d = 1.41$, 95% CI = [0.82–1.99]). Results for the individual stimuli are presented in Supplementary Fig. 3.

In experiment 3, we found attenuated performance levels when complex stimuli were associated with low reward probabilities compared to complex stimuli associated with high reward probabilities already when no delay was imposed. Here, high reward stimuli received more correct responses compared to low reward stimuli (high reward = 96.1% correct responses; low reward = 88.8% correct responses, $t = 11.37$, $p < 0.001$, $d = 1.58$, 95% CI = [1.28–1.87], see Supplementary Table 4 for all descriptive data). These results suggest that reward probabilities affected the general ability to associate stimuli with the adequate choice clearly influencing the choice behavior. However, different reward levels between the complex and simple stimulus sets do not account for the difference in the effect of delay performance between these two sets.

In conclusion, we consistently showed in three behavioral experiments that stimulus complexity had a strong impact on working memory performance. For stimuli comprising luminance, spatial, and color information, the introduction of a memory delay only had small, if any effects on maintenance. If simple stimuli, comprising only luminance information, were presented, performance was significantly diminished following a memory delay. Figure 2d provides an overview across all three experiments.

**Neural recordings in the NCL reveal coding differences for simple and complex stimuli.** Next, we wanted to investigate neural representations that correlate with the differences in working memory performance that we observed in our behavioral experiments. To this end, we recorded single neuron activity in the NCL while the animals performed the delayed condition of experiment 3 as this condition comprised all relevant manipulations that affect the choice behavior, i.e. working memory for stimuli of different complexity, as well as the manipulation of value. 104 neurons from 34 behavioral sessions of three pigeons (see Supplementary Table 1 for a detailed description) were recorded from the NCL (for histological electrode track reconstruction see Supplementary Fig. 4). Performance rates were highly comparable between the recording sessions and the purely behavioral sessions from the delayed condition of experiment 3 (see Supplementary Fig. 5).

**Different neural activation pattern during a visual working memory task.** At first, we were interested in the neural representation of task parameters. Particularly, we investigated if there were systematic differences between the two stimulus classes during working memory. The stimulus classes were analyzed separately as we identified major performance differences between them in the preceding behavioral experiments. We analyzed trials with complex and simple stimuli independently using a sliding window ANOVA (250 ms bin size) with the factors "response" (two levels: left or right) and "value" (levels high vs. low reward probability for complex stimuli or high vs. low contrast for simple stimuli). For each factor, we computed the effect size $\eta_p^2$ in each given bin of the trial which reflects the percent of explained variance (PEV; see Fig. 3a for illustration of these factors). We classified each neuron into four categories based on the results of the ANOVA and the ensuing effect sizes (see Methods *Task related activity – Cell classification* for details):

1. **Choice-related activity**: A difference in the neural response between left and right choices, resulting in a main effect of factor "response" ($\eta_p^2 > 0.03$ in two consecutive bins).
2. **Value-related activity**: A difference in the neural response between high and low rewarded stimuli or high and low contrast stimuli, resulting in a main effect of factor "value" ($\eta_p^2 > 0.03$ in two consecutive bins).
3. **Stimulus-related activity**: If a neural response was related to one specific stimulus, an idiosyncratic result of the ANOVA was observed. Only in this specific case, value, choice and interaction effects occurred at the same time ($\eta_p^2 > 0.03$ in two consecutive bins for the factors "value", "response" and their "interaction" simultaneously)[35].
4. **Interaction activity:** This type of representation arose whenever a single neuron represented more than one single stimulus or when value-related activity switched to a choice-based pattern within the very same cell, resulting in an interaction between the factors "value" and "response" ($\eta_p^2 > 0.03$ in two consecutive bins).

We quantified the difference in effect size between stimulus classes during the sample and delay phase. This was performed separately for the different task periods, i.e. the sample phase, 1st second of the delay, 2nd second of the delay and 3rd second of the delay.

**Choice-related activity**. The first activity type we investigated was choice-related activity—a neural response associated with the side of the animals' response. To verify that this activity was indeed associated with the animal's subsequent decision, we compared correct and incorrect trials in a supplementary analysis (Supplementary Figs. 6, 7). An example neuron demonstrating choice-related activity is depicted in Fig. 3b (raster) and 3c (spike density function; SDF). For simple stimuli, this neuron did not show any relevant activity changes during the sample phase whereas it exhibited stimulus-related activity for complex stimuli. During the delay phase, this neuron differentiated between left and right decisions both for complex and simple stimuli.

The results for choice-related activity of each individual neuron are presented in Fig. 3d (left: complex stimuli, right: simple stimuli). Choice-related activity in the NCL was found extensively, but its extent differed notably between the stimulus classes. Comparing the average effect of choice-related activity between both stimulus classes during the sample phase revealed significantly less activity for complex compared to simple stimuli (mean effect complex = 0.001, mean effect simple = 0.008; $t = 2.59$, $p = 0.010$, $d = 0.36$, 95% CI = [0.08–0.64]). Across the delay phase, effect sizes of choice-related activity did not differ between the different stimulus sets at any point in time (all $p$s > 0.05).

**Value-related activity**. The second aspect captured by our analysis was value-related activity, i.e. activity differences between high and low reward complex stimuli or differences between high and low contrast simple stimuli. An example neuron representing value differences for complex stimuli is shown in Fig. 4b (raster plot) and 4c (SDF). This neuron differentiated significantly between stimuli of high and low reward probability during the sample phase (value effect size is highlighted in red; Fig. 4c).

The results for value-related activity of each neuron are presented in Fig. 4d. Value coding was most prevalent during the sample phase and could exclusively be found for complex stimuli. Comparing the average effect of value-related activity between both stimulus classes during the sample phase revealed significantly more value-related activity for complex compared to simple stimuli (mean effect complex = 0.005, mean effect simple < 0.001, $t = 2.22$, $p = 0.028$, $d = 0.31$, 95% CI = [0.03–0.58]). This was also true for the first second of the delay phase (mean effect complex = 0.003, mean effect simple < 0.001; $t = 2.56$, $p = 0.011$, $d = 0.35$, 95% CI = [0.08–0.63]). Value-related activity was absent further on during the task (see Fig. 4d and Table 1). Thus, no differences in value-related activity were found during these task periods.

**Stimulus-related activity**. The third aspect we investigated was stimulus-related activity, i.e. whenever a neuron represented one stimulus in particular. An example neuron of this kind can be found in Fig. 5b (raster plot) and 5c (SDF). This neuron was selectively activated during the presentation of one particular complex stimulus exclusively during the sample phase (stimulus effect size is highlighted in yellow; Fig. 5c).

The results for stimulus-related activity of each recorded neuron are presented in Fig. 5d (left: complex stimuli, right: simple stimuli). Stimulus-related activity was most prevalent

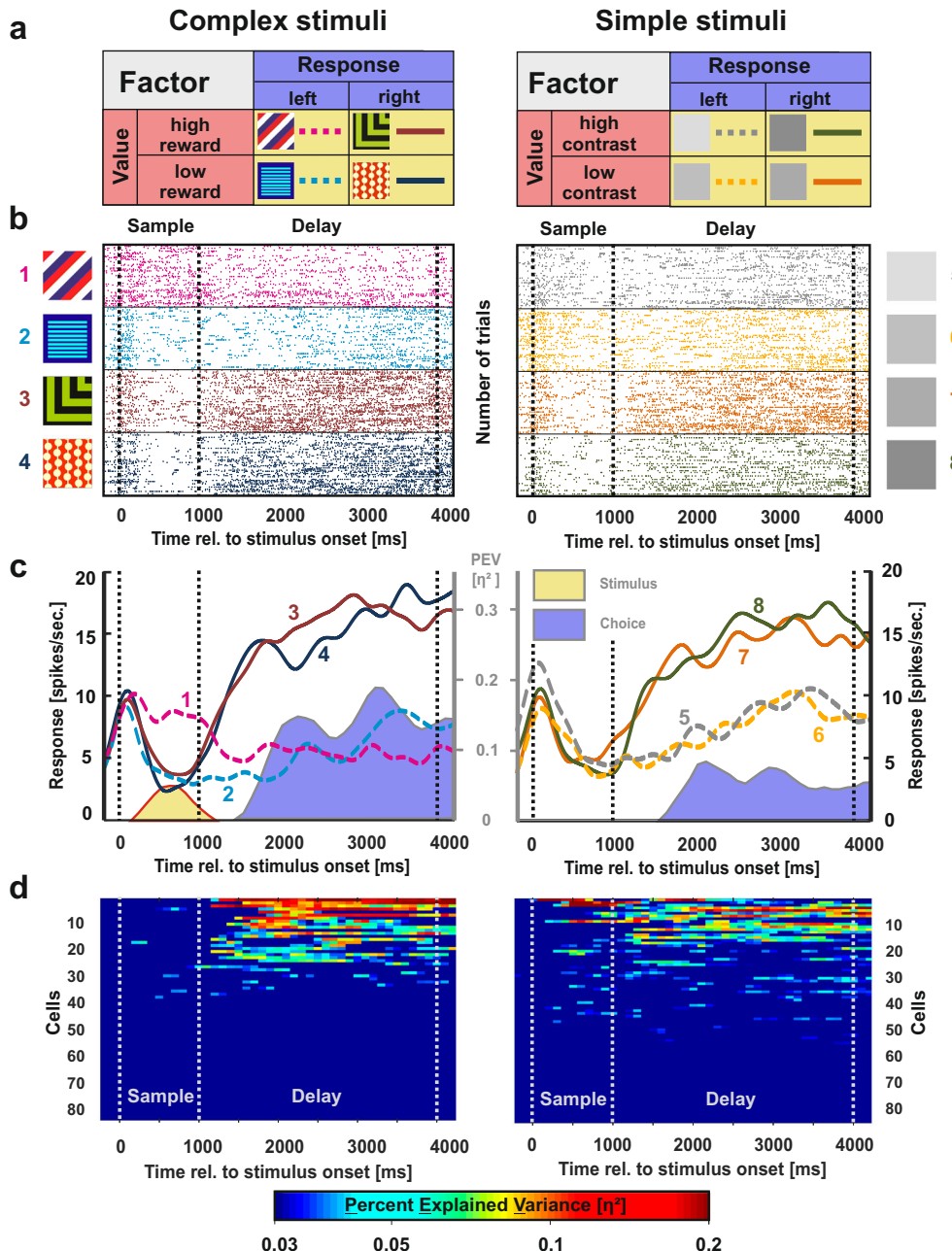

**Fig. 3 Example neuron and population response for choice-related activity in the NCL. a** The rationale of the ANOVA for the electrophysiological data. Stimuli were either associated with a left (dashed lines) or right choice (solid lines, factor choice) or with the higher or lower net outcome due to different reward probabilities in the complex stimulus class (left panel) or discrepancies in contrast for the simple stimulus class (right panel). The colors in the figure represent the different activity patterns of NCL neurons (stimulus = yellow, value = red, choice = blue). **b** Raster plot of an example neuron demonstrating choice-related activity. For both complex and simple stimuli, this neuron differentiated between stimuli associated with a left and right choice during the delay phase. For complex stimuli, the neuron furthermore showed stimulus-related activity during the sample phase (left). **c** Spike density function of the example neuron shown in **b**. The numbers of the line plots correspond with one of the eight experimental stimuli. The PEV by the relevant coding type is presented behind the SDF as shaded area in the respective color of the activity pattern (stimulus = yellow, value = red, choice = blue; cf. **a**). Corresponding values are scaled on the secondary axis depicted in gray. **d** Population response for choice-related activity of all individual neurons for trials in which complex (left panel) and simple stimuli (right panel) were presented. Cells that did not exhibit any activity according to the ANOVA during any phase ($n = 22$ cells) were excluded from the plot. Only above threshold firing ($\eta_p^2 > 0.03$) is shown.

during the sample phase and could almost exclusively be found for complex stimuli. We then compared the average effect of stimulus-related activity between both stimulus classes during the sample phase. Here, there was stronger stimulus-related activity for complex compared to simple stimuli (mean effect complex = 0.009, mean effect simple = 0.001; $t = 3.52$, $p = 0.005$, $d = 0.49$, 95% CI = [0.21–0.77]). During the delay phase, stimulus-related

activity was virtually absent for both stimulus classes, consequently there were no differences in stimulus-related activity between the classes.

We were also interested whether the number of stimulus-representing neurons was driven by reward contingencies, i.e. if neurons represented stimuli associated with high reward more often. Of the 20 cells modulated by the complex stimuli, 12 cells

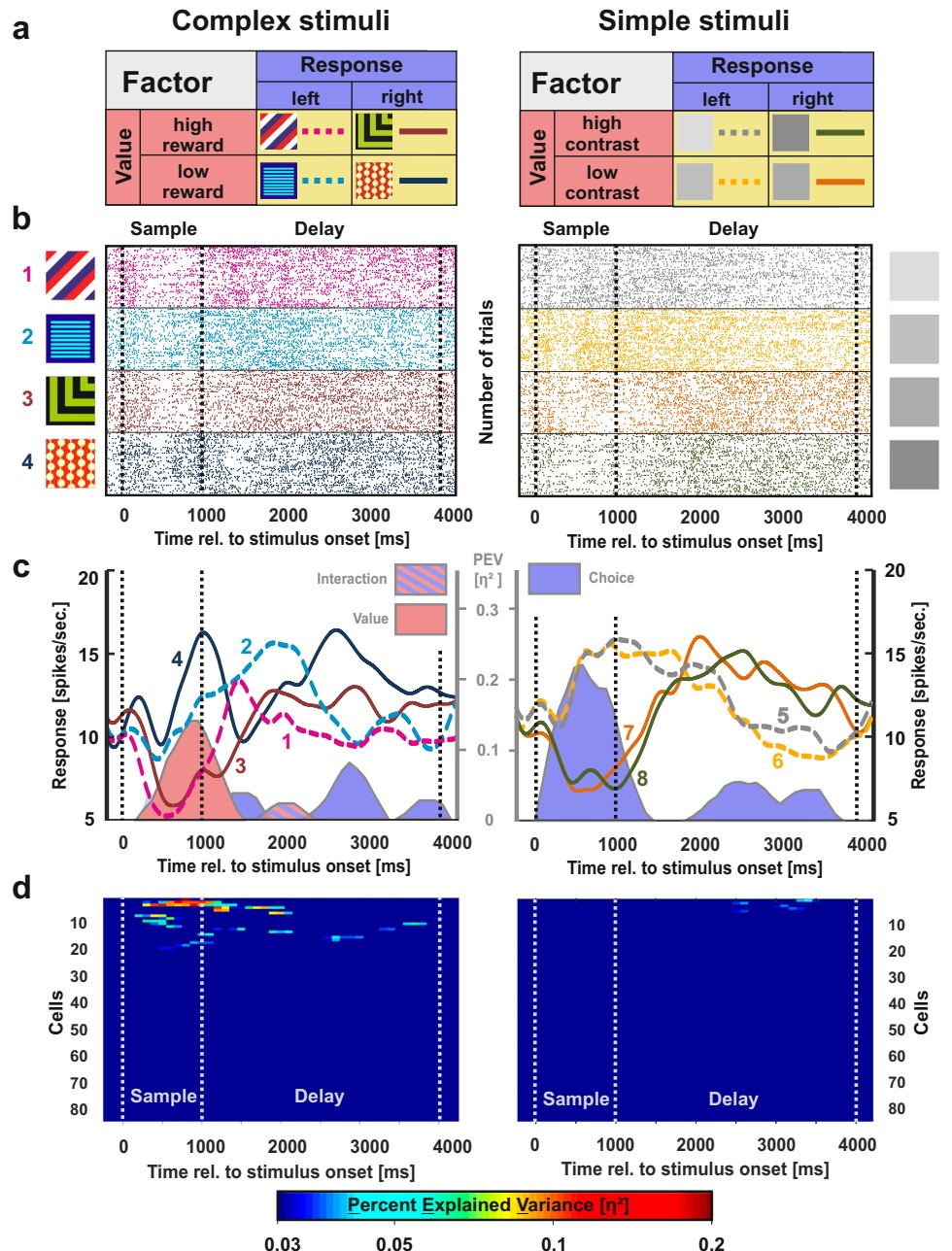

**Fig. 4 Example neuron and population response for value-related activity in the NCL. a** The rationale of the ANOVA as described in Fig. 3. **b** Raster plot of a value representing neuron. Dashed vertical lines indicate borders of the phases. This neuron differentiated between low and high reward stimuli during the sample phase for complex stimuli (left). For simple stimuli, the upcoming choice was represented already during the sample phase (right). During the delay, this neuron demonstrated a choice code for complex and simple stimuli alike. **c** Spike density function of the example neuron shown in **b**. The PEV by the relevant activity type is presented behind the SDF as shaded area in the respective color of the activity pattern (stimulus = yellow, value = red, choice = blue; cf. **a**). Corresponding values are scaled on the secondary axis depicted in gray. **d** Population response for value-related activity of all individual neurons for trials in which complex (left panel) and simple stimuli (right panel) were presented.

coded for the high rewarded stimuli whereas eight cells were active in the presence of low rewarded stimuli. No significant difference could be detected between them ($\chi^2_{(1)} = 0.8$, $p > 0.250$) indicating that reward probability did not drive stimulus-related activity in the NCL population.

**The interaction of value and choice.** The fourth aspect we analyzed was the interaction between the factors value and choice. This type of representations arose whenever a single neuron represented more than one single stimulus or when value-related

activity switched to a choice-based pattern within the very same cell. An example neuron representing two stimuli simultaneously resulting in a significant interaction effect is shown in Supplementary Fig. 8b (raster) and 8c (SDF). Interaction effects were most prevalent during the sample phase and could almost exclusively be found for complex stimuli. Comparing the average effect of interactions of value and choice between both stimulus classes during the sample phase revealed stronger activity in the interaction category for complex compared to simple stimuli (mean effect complex = 0.002, mean effect simple < 0.001; $t = 2.97$, $p = 0.003$, $d = 0.41$, 95% CI = [0.14–0.69]). This was

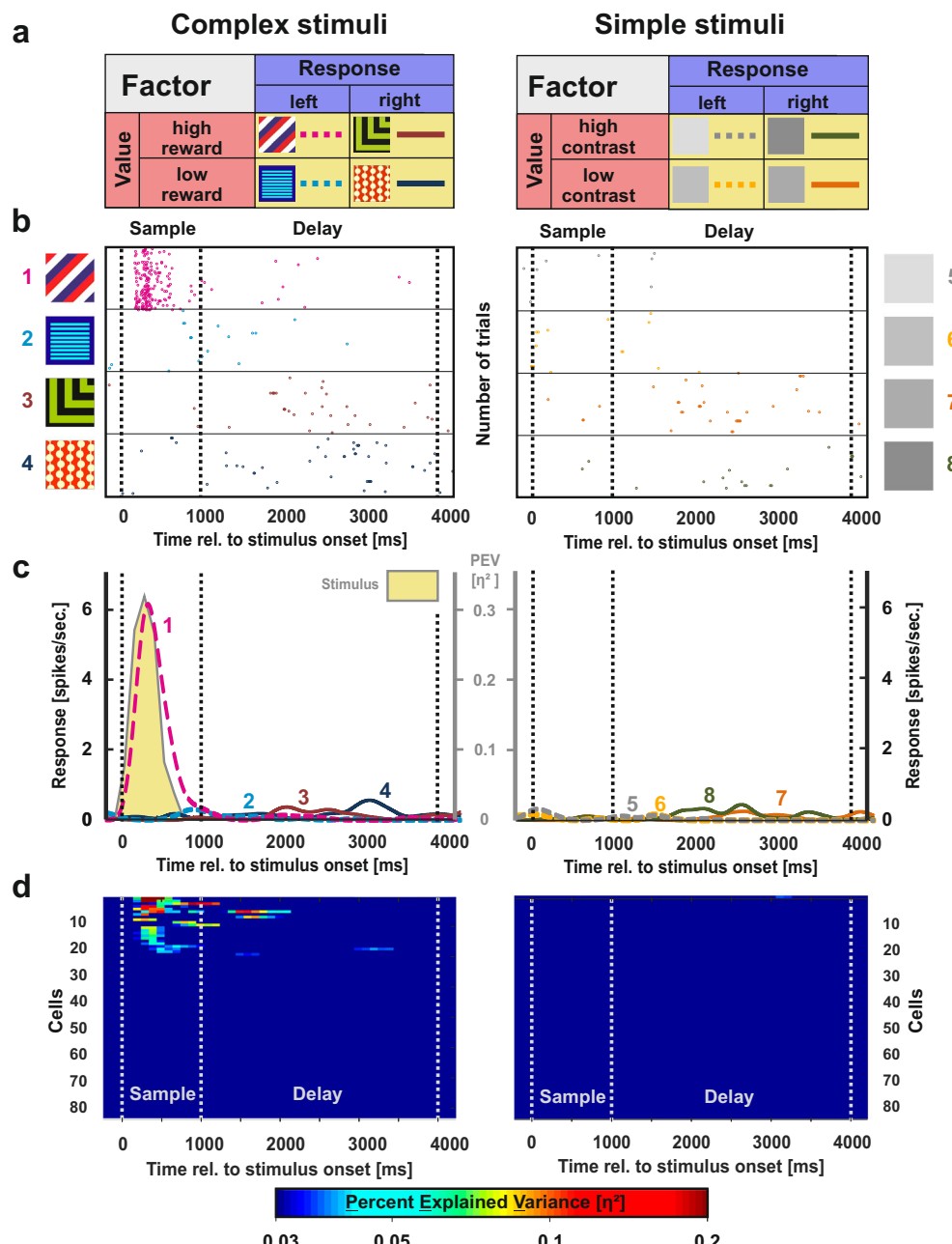

**Fig. 5 Example neuron and population response for stimulus-related activity in the NCL. a** The rationale of the ANOVA as described in Fig. 3. **b** Raster plot of a stimulus representing example neuron. This neuron specifically responded to one of the high rewarded complex stimuli during the sample phase (left) and to no stimulus from the simple stimuli (right). **c** Spike density function (SDF) of the example neuron shown in **b**. The percent explained variance (PEV) by the relevant activation type is presented behind the SDF as shaded area in the respective color of the activity pattern (stimulus = yellow, value = red, choice = blue; cf. **a**). Corresponding values are scaled on the secondary axis depicted in gray. **d** Population response for stimulus-related activity of all individual neurons for complex (left panel) and simple stimuli trials (right panel).

also true for the first second of the delay phase (mean effect complex = 0.002, mean effect simple < 0.001; $t = 2.81$, $p = 0.005$, $d = 0.39$, 95% CI = [0.11–0.67]).

**Summarizing comparison of neural activation pattern for simple and complex stimuli.** Table 1 summarizes the above mentioned cell categories (i.e. choice-, value-, stimulus-, and interaction activity) across the experimental phases (i.e. sample phase and delay). To provide a comparison of activity patterns during the sample phase (1 s) and the delay phase (3 s), we divided the delay phase into three time windows of 1 s duration.

To identify if the number of classified cells were above what was to be expected by chance, we randomly permuted spike counts to determine how often neurons were classified into each category based on chance alone (see Methods *Chance-based quantity of active cells per category* for detail, Supplementary Table 5 for results). Finally, we used Fisher's exact test to identify whether the classification result was significantly different from the chance-based prediction. Since for each stimulus class 20 individual phases had to be tested (phases*activity type; cf. Table 1), we used a Bonferroni corrected threshold of $p < 0.0025$. Note that individual neurons could potentially be classified into multiple categories, as the categories were not mutually exclusive.

**Table 1 Distribution of active cells per category (choice, value, stimulus, and interaction activity) across the sample and delay phase for both the complex and simple stimuli.**

| Stimulus Set | Complex Stimuli | | | | |
|---|---|---|---|---|---|
| | Choice | Value | Stimulus | Interaction | Cell Total |
| Sample Phase | 8 (7.69%) | 13 (12.5%)* | 20 (19.23%)*** | 13 (12.5%)* | 48 (46.15%)*** |
| Delay 1st second | 34 (32.69%)*** | 10 (9.62%) | 5 (4.81%) | 15 (14.42%)** | 39 (37.5%)*** |
| Delay 2nd second | 24 (23.08%)*** | 4 (3.85%) | 2 (1.92%) | 5 (4.81%) | 33 (31.73%)*** |
| Delay 3rd second | 24 (23.08%)*** | 3 (2.88%) | 2 (1.92%) | 1 (0.96%) | 30 (28.85%)*** |
| | Simple Stimuli | | | | |
| | Choice | Value | Stimulus | Interaction | Cell Total |
| Sample Phase | 21 (20.19%)*** | 0 (0%) | 1 (0.96%) | 1 (0.96%) | 23 (22.12%)*** |
| Delay 1st second | 36 (34.62%)*** | 0 (0%) | 0 (0%) | 1 (0.96%) | 37 (35.58%)*** |
| Delay 2nd second | 29 (27.88%)*** | 3 (2.88%) | 0 (0%) | 3 (2.88%) | 34 (32.69%)*** |
| Delay 3rd second | 29 (27.88%)*** | 4 (3.85%) | 1 (0.96%) | 2 (1.92%) | 33 (31.73%)*** |

The asterisks indicate if the number of observed neurons is higher than expected by chance (Fisher's exact test). Please note that the categories are not mutually exclusive as one neuron can for example exhibit both stimulus coding as well as value coding during the sample phase. For that reason, the cell total is not necessarily equal to the sum of all active neurons. *P*-value thresholds are indicated by asterisks and represent the Bonferroni corrected values.
*$p < 0.05$; **$p < 0.01$; ***$p < 0.001$

During the sample phase, we found 7.69% choice coding neurons for complex stimuli. The number of cells was not significantly different from chance level. However, 12.5% of the recorded neurons showed value-related activity in response to the complex stimuli. In addition, 19.23% of the recorded neurons were responsive to one specific complex stimulus and 12.5% of the recorded cells displayed interaction activity between the factors choice and value. All of these responses were significantly different from chance (Table 1). The presentation of simple stimuli during the sample phase resulted in an opposite activity pattern: 20.19% of the recorded neurons displayed choice-related activity. An amount of cells that was clearly above chance. However, the NCL population did not demonstrate significant numbers of neurons representing value- (0%) or stimulus-related (0.96%) activity nor above chance interaction activity between the factors choice and value (0.96%).

Throughout the delay phase, we found a significant fraction of neurons in the NCL population that represented the upcoming choice for complex stimuli (1st second: 32.69%; 2nd second: 23.08%; 3rd second: 23.08%). Further, 14.42% of the recorded neurons showed a significant interaction activity between the factors choice and value for complex stimuli in the first second of the delay period. For simple stimuli, choice-related activity was displayed throughout the delay period exclusively (1st second: 34.62%; 2nd second: 27.88%; 3rd second: 27.88%).

Thus, when analyzing neuronal activation pattern during different phases within a trial, a clear distinction between simple and complex stimuli becomes apparent especially during the sample period: While the presentation of complex stimuli resulted in stimulus representation, value representation and the interaction of both response types, solely choice activity was found when simple stimuli were presented.

**Temporal dynamics of neural activation schemes.** So far, our analysis was based on the population response for single task parameters such as choice or value but it did not investigate if individual neurons dynamically represent different task parameters, a phenomenon known as neural multiplexing[36,37]. To investigate whether NCL neurons were encoding only one task aspect (i.e. solely choice, value, stimulus, or interaction activity) or were engaged in the representation of multiple aspects at different stages of the task, we tracked task-related activity patterns of individual neurons over time and categorized them accordingly.

As evident from the example neurons shown in Figs. 3b/c, 4b/c, individual neurons were not limited to one type of activation pattern but employed strong multiplexing resulting in dynamic changes from the sample to the delay phase when presented with complex stimuli. For simple stimuli, these neurons exclusively represented choice-related activity.

Based on the results of the ANOVA, we created four post hoc categories to classify individual neurons according to their firing pattern dynamics across the sample and delay phase:

1. **Early choice:** The upcoming choice was represented already during the sample phase. No stimulus, value or interaction activity is observed at any point during the task. An example neuron is depicted in Fig. 6a1 (raster) and 6a2 (SDF).
2. **Late choice:** The upcoming choice was represented after the sample phase had ended. No stimulus, value or interaction activity is observed at any point during the task. An example neuron is depicted in Fig. 6b1 (raster) and 6b2 (SDF).
3. **Stimulus - No choice:** Neurons that demonstrated stimulus, value or interaction activity, but no choice-related activity. An example neuron is depicted in Fig. 6c1 (raster) and 6c2 (SDF).
4. **Stimulus - Choice:** Neurons that demonstrated stimulus, value or interaction activity initially during the trial and then dynamically switched to a choice representation later in the trial. An example neuron is depicted in Fig. 6d1 (raster) and 6d2 (SDF).

The two stimulus classes differed with regard to the number of "early choice" and "late choice" neurons. Significantly fewer neurons demonstrated early choice-related activity with complex than with simple stimuli (complex: six neurons, simple: 21 neurons, $\chi^2_{(1)} = 9.58$, $p = 0.002$). The same was true for "late choice" representations (complex: 13 neurons, simple: 26 neurons, $\chi^2_{(1)} = 5.33$, $p = 0.021$). Thus, representations of choice were significantly more prevalent for simple stimuli. Interestingly, "early choice" neurons demonstrated significantly higher effects of choice-related activity at the beginning than at the end of the delay for simple stimuli ($t_{(9)} = 8.70$, $p < 0.001$, see Fig. 6a3) indicating that the strength of the choice-related activity decayed after an early onset. Activity levels of "late choice" neurons did not significantly decay throughout the delay (Fig. 6b3). We then compared the number of active cells for the "stimulus - no choice" and the "stimulus - choice" category between the two

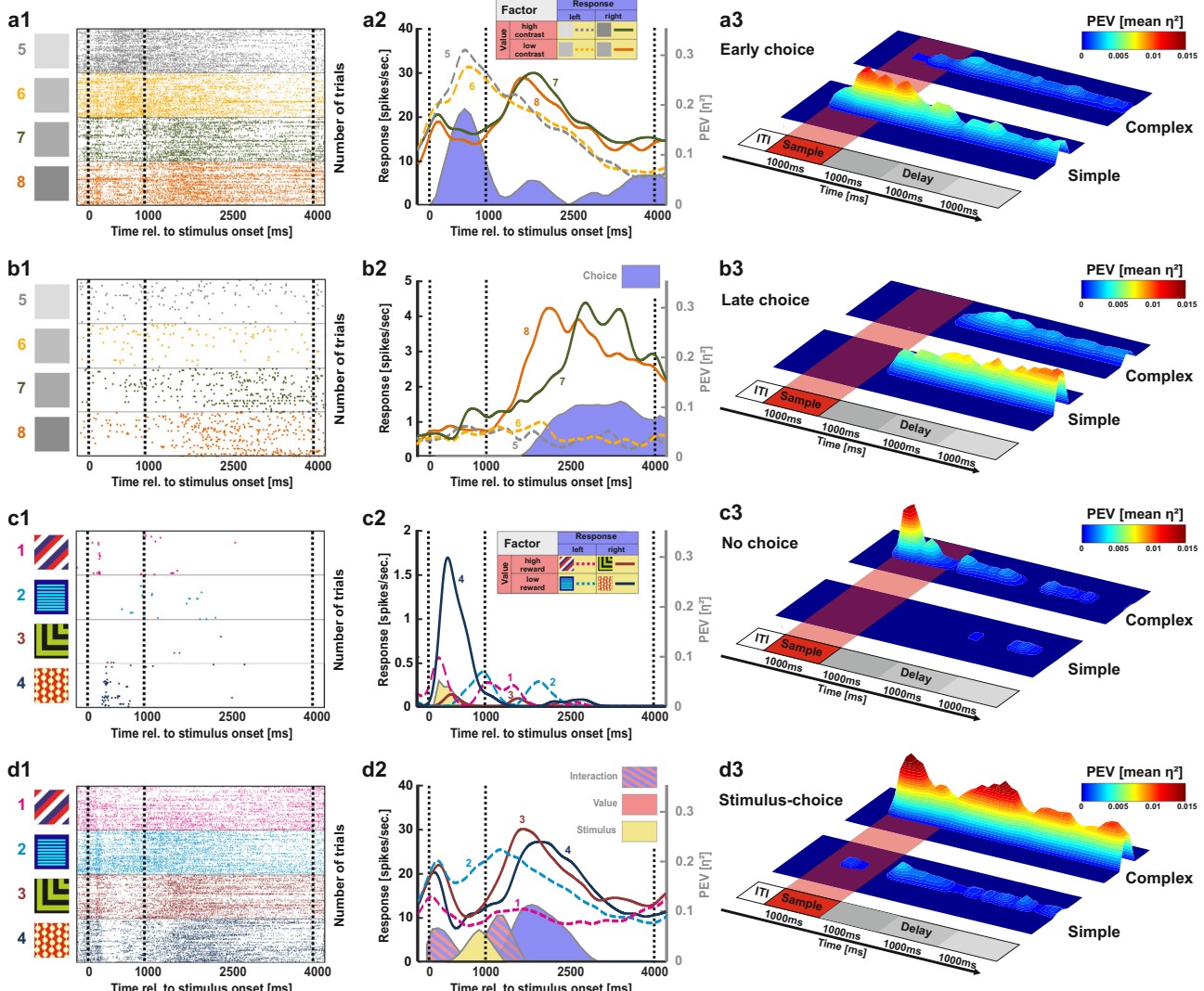

**Fig. 6 Neuronal coding schemes in the NCL. a** The first category comprised neurons that encoded the upcoming decision already during the sample phase ("early choice"). a1 Raster plot of an example neuron that differentiated between left and right choices during sample and delay phase for the simple stimuli. a2 According SDF and effect sizes of the example neuron. a3 Population average for all neurons demonstrating "early choice" representations for both the simple and complex stimuli. **b** The second category comprised neurons that represented the upcoming decision from delay phase onset onwards ("late choice"). b1 Raster plot of an example neuron that differentiated between left and right choices during the delay for the simple stimuli. b2 According SDF and effect sizes of the example neuron. b3 Population average for all neurons demonstrating "late choice" representations for both the simple and complex stimuli. **c** The third category comprised neurons that encoded solely task parameters not associated with the upcoming choice, i.e. stimulus, value and/or interaction activity during either the sample or the delay phase ("stimulus - no choice"). c1 Raster plot of an example neuron that demonstrated stimulus-related activity during the sample phase for one complex stimulus. c2 According SDF and effect sizes of the example neuron. c3 Population average for all neurons demonstrating "stimulus - no choice" representations for both the simple and complex stimuli. **d** The fourth category comprised neurons that initially encoded representations not associated with the upcoming choice, i.e. stimulus, value and/or interaction activity, but later switched to choice-related activity patterns ("stimulus - choice"). d1 Raster plot of an example neuron that demonstrated stimulus-related and interaction activity during the sample and early delay phase for complex stimuli. The neuron then dynamically switched to choice-related activity during the delay phase. d2 According SDF and effect sizes of the example neuron. d3 Population average for all neurons demonstrating "stimulus - choice" representations for both the simple and complex stimuli.

stimulus classes. Here, 33 cells demonstrated "stimulus - no choice" representations when confronted with complex stimuli compared to four cells when confronted with simple stimuli ($\chi^2_{(1)} = 9.58$, $p < 0.001$, see Fig. 6c3 for population results). "Stimulus - choice" neurons were also more prevalent when the animals were confronted with complex compared to simple stimuli (complex: 16 neurons, simple: five neurons, $\chi^2_{(1)} = 6.41$, $p = 0.011$). In contrast to "early choice" neurons, activity levels did not decay from early to late stages of the delay in neurons employing such a stimulus-choice representation when complex stimuli were presented ($t_{(9)} = 0.60$, $p = 0.561$, Fig. 6d3).

Overall, NCL neurons show multiplexing by dynamically switching from stimulus, value or interaction activity to choice-related activity. However, this coding scheme was exclusively found when complex stimuli were presented. In this case, the subsequent choice component was maintained at a stable level throughout the entire delay period. For simple stimuli, neuronal multiplexing was entirely absent since neurons did not represent stimulus, value, or interaction activity at all. Instead, choice-related activity emerged already during the presentation of simple stimuli. Population activity of these "early choice" neurons subsequently degraded over the delay phase.

## Discussion

In the present study, we investigated the effects of stimulus-complexity on working memory performance using both behavioral and neurophysiological approaches in pigeons. In line with our hypothesis, complex stimuli that comprised luminance, spatial, and color information could be maintained better in working memory compared to simple stimuli void of visual features except for luminance. This distinction was associated with a clear difference between the neural representations of these stimulus classes. NCL neurons represented only idiosyncratic features of complex stimuli such as their value or physical identity during the sample phase. Only later, during the delay period, the upcoming choice was represented. Importantly, the representation of different task parameters (e.g. stimulus and choice) occurred within single neurons illustrating neuronal multiplexing. Simple stimuli, on the other hand, were never represented in NCL. Instead, we only recorded signals reflecting the upcoming choice even while the images were still visible to the animal. Since we exclusively discovered choice-related activity for simple stimuli, no neuronal multiplexing could be detected.

The causal involvement of the NCL in working memory tasks was initially shown with lesion studies in pigeons[32,38]. Subsequent electrophysiological recordings in the pigeon NCL during working memory tasks showed enhanced delay activity, especially in trials in which a motor response was required and reward was delivered[33,39]. In accordance with these studies, we found neuronal representations of value, choice, and their interactions for complex stimuli. For both stimulus classes, we found responses during the delay phase that signaled the upcoming choice. Studies in the NCL of crows further showed stimulus selective activity during the presentation period. This activity was subsequently maintained during the delay period[31,40]. Importantly, in the study by Veit et al.[31], half of the recorded neurons showed a differential response pattern during the sample and delay phase. Especially these complex, diverse, and mixed selective responses are in full agreement with our data (for a further discussion on mixed selective responses see below). In these studies, only complex pictures were used as stimuli and thus, the impact of stimulus complexity has not been investigated with electrophysiological methods. However, in a behavioral study, the interfering impact on working memory was shown to be small for simple distractor stimuli and more severe for complex pictures, hinting towards differential processing of both stimulus classes[41]. To systematically investigate interfering effects of the different stimulus classes, an experiment as performed by Wright et al.[42] would be needed. Testing the serial position function of the different stimulus classes under diverse delay conditions would shed further light on differential encoding and the resulting interference pattern for each stimulus class. Additionally, increasing the number of items would inform about the working memory capacities for stimuli of different visual quality.

The idea that stimulus features impact working memory performance is rather recent. Results that some hues are memorized more precisely than others or that cardinal directions are easier to memorize than other orientations can probably be explained by categorization that could depend on stimulus statistics[17–19,43]. These findings have usually been conceptualized in the scope of computational models investigating capacity limits[44,45], noise accumulation[46], or discrete memory storage[20,47]. Importantly, all these frameworks indicate a clear relation between the identity of a specific visual stimulus and memory performance that is grounded in a systematic difference of the stimulus' neural representation. Our results support this notion as the enhanced performance for complex compared to the simple stimuli could be linked to a differential recruitment of neural populations for the two stimulus classes. Complex stimuli offered a large number of idiosyncratic visual features that could have aided in bringing the working memory representation into a stable state. A potential explanation how this occurred on the neural level could relate to different activity schemes depending on the presented stimulus class: we exclusively found neuronal multiplexing when complex stimuli were presented. Neural multiplexing refers to a neuron's capacity to be activated by multiple different task relevant parameters as opposed to highly specialized neurons[48] likely due to being embedded in diverse neural ensembles. Here, individual neurons represented a multitude of task parameters not only at different points during the trial but also simultaneously with regards to the interaction between value and choice. Thus, representations of complex stimuli were of higher dimensionality in the NCL compared to those of simple stimuli.

High dimensional coding by neurons has been suggested to be associated with a facilitated read-out by downstream neurons[49]. Since active maintenance in working memory relies on bursting populations rather than single neurons carrying the information, especially across long delays[6], it seems feasible that a facilitated read-out through high dimensional representations aids the transmission of said information. Furthermore, both in rodents and primates, studies investigating high dimensional coding properties of (pre-)frontal cortices during the execution of higher cognitive tasks have demonstrated that this dimensionality collapses in error trials indicating that it is subserving goal-directed behavior[50,51]. Overall, our electrophysiological findings therefore suggest that high dimensional representations are tied to the presented stimulus material. A recent study has furthermore demonstrated that attention and working memory share a common neural basis in the PFC[15] hinting at the possibility that the presence of manifold stimulus features increased attention towards the stimuli enhancing the neural representation in working memory.

A consequence of the differential activity pattern during the stimulus presentation is the concomitant difference in the onset of neural activity reflecting the upcoming choice. If the switch from stimulus-related to choice-related activity is taken as an indicator of decision time[52–54], the decision happened significantly later for complex stimuli (almost exclusively after delay onset) shortening active maintenance of the choice-related activity in working memory. Longer maintenance times have been demonstrated to result in a decay of the memory trace, which is hypothesized to be a consequence from stochastic neural noise causing the degradation of the stored memory over maintenance time[55,56]. For simple stimuli, we observed no stimulus- or value-related activity during the sample phase. Rather, choice-related activity emerged much earlier already during stimulus presentation and as a direct result, maintenance time for the upcoming choice was prolonged. This could have resulted in increased trace decay compared to the complex stimuli since there was an increased opportunity for noise to accumulate and for interference to occur.

While our study provides interesting insights how stimulus complexity affects working memory performance on the behavioral level and about the neural correlates associated with this phenomenon, our study is subject to a few limitations that should be acknowledged. On the one hand, we did not record from neural structures other than the NCL in the task and thus our study does not inform how specific the observed neural responses are to the NCL. Future studies should be conducted in the same behavioral framework while recording from other brain structures to identify if the differentiation between complex and simple stimuli is selectively represented in the NCL. On the other hand, we cannot rule out that the observed multiplexing was due to insufficient isolation of single units. However, given our strict sorting criteria and the observation of multiplexing in a large

number of neurons in the NCL population for complex stimulus trials, we deem it unlikely that this strongly affected our dataset.

In our study, we provide evidence that stimulus material is influencing working memory. We found that high stimulus-complexity of a visual stimulus positively affects working memory performance compared to low stimulus-complexity. Our complementary neuronal data indicate that the neural representations associated with our two stimulus classes differed substantially. For complex stimuli, we found high dimensional multiplex activity patterns as well as a delayed onset of choice-related activity in the NCL population. For simple stimuli, choice-related activity was immediately present and the population activity was markedly low dimensional. These differences in representation could provide an account for the behavioral differences observed. Since working memory research has largely focused on memory load rather than the stimulus material, our study highlights the importance to carefully select the stimulus material.

## Methods

**Experimental model and subject details.** Subjects were treated in accordance with the German guidelines for the care and use of animals in science and all experimental procedures were approved by a national ethics committee of the State of North Rhine-Westphalia, Germany (LANUV) and were in agreement with the Directive 2010/63/EU of the European Parliament and of the Council of 22 September 2010 concerning the care and use of animals for experimental purposes. We have complied with all relevant ethical regulations for animal testing. 27 adult homing pigeons (*Columba livia*) of unknown sex were used in this study. The pigeons were between 1 and 6 years of age. The birds used as experimental subjects were obtained from local breeders. Of these, three birds were employed in electrophysiological experiments whereas 24 animals solely took part in behavioral studies (experiment 1: 11 pigeons, experiment 2: 8 pigeons and experiment 3: 17 pigeons; animals were partially reused between experiments). Birds were accommodated individually in wire-mesh cages within a colony room (12 h light/dark cycle; beginning at 8:00 h) where they had ad libitum access to water and grit. On testing days, food access was restricted and the animals were maintained between 80–90% of their free-feeding body weight.

**Apparatus.** Testing was conducted in a custom-built operant chamber (33 × 34 × 34 cm[57]). The rear wall of the chamber featured three horizontally aligned rectangular translucent choice keys (4 × 4 cm wide) located above a food hopper. An LCD screen was mounted against the rear wall for the presentation of stimuli at the choice key locations. Pecks to the choice-keys were immediately followed by auditory feedback. The chamber was illuminated by two lights at the top of the chamber with an additional feeder light affixed on top of the food hopper. The chamber was situated in a sound-attenuating cubicle and all experimental sessions were conducted with a constant presentation of white noise (~60 dB) to prevent external noise from distracting the animals during the task. Hardware was controlled by a custom written MATLAB code (2018a, The Mathworks, Natick, MA, USA)[58].

**Stimuli.** The stimulus set encompassed eight stimuli (Fig. 1b), consisting of four shaped multicolored images (complex stimuli) and four gray stimuli of different luminance (simple stimuli). The gray stimuli were subdivided into two bright stimuli (86.3% white and 98% white) and two dark stimuli (43.1% white and 54.9% white). Due to the number of luminance increments between the respective stimulus and the category boundary (70.6% white) they were either easy or hard to discriminate.

While the complex stimuli provide at face value more feature information compared to the simple stimuli, we also verified the images' visual information both in the spatial and color dimension. To estimate the spatial information of the stimulus classes, we used a bank of Gabor filters. These filters are used for edge detection and luminance identification in image processing and they have widely been used as a model of simple cells in the primary visual cortex[59–62]. The 68 distinct Gabor filters were divided into four orientations θ (0 deg, 45 deg, 90 deg and 135 deg) and 17 sizes s ∈ {1,…, 17} using the following formula:

$$F(x, y) = N\left[\frac{1}{N} + H(G(x, y) - 1)\right]G(x, y) \quad (1)$$

Here, x and y are the coordinates within the receptive field of the Gabor filters.

$$N = \left(1 \Big/ \int dx \int dy G(x, y)\right) - 1 \quad (2)$$

is the normalization factor that ensures an upper bound of 1 for Gabor convolutions.

$$H(x) = 1 \text{ for } x \geq 0 \quad (3)$$

is the Heaviside step function. The Gabor function is defined as follows:

$$\exp\left(-\frac{x'^2 + \gamma^2 y'^2}{2\sigma^2}\right)\cos\left(2\pi\frac{x'}{\lambda}\right) \quad (4)$$

where x' and y' are defined as follows:

$$x' = x\cos(\theta) + y\sin(\theta) \quad (5)$$

$$y' = -x\sin(\theta) + y\cos(\theta) \quad (6)$$

The width of the Gabor filter based on the size of the filter,

$$\sigma = 0.0036 s^2 + 0.35 s + 0.18 \quad (7)$$

And the frequency of the width,

$$\lambda = \frac{\sigma}{0.8} \quad (8)$$

The Gabor filters are shown in Supplementary Fig. 9. We calculated the convolution of the filters with each image used in the study. The average absolute values of the pixels in the resulting convoluted images were then arranged into a vector representing the image. We then extracted the median feature density of each stimulus (Supplementary Fig. 10a).

To estimate the color information of the stimulus classes, we measured the luminance and chromaticity using a Konica Minolta Chroma Meter (CS-150) and averaged the result over 10 repeated measurements (Supplementary Fig. 10b).

**Behavioral paradigm.** Subjects were trained in a paired association task that was divided into three distinct experimental conditions (Fig. 1): in all experiments, a placeholder initialization stimulus appeared on the center key for up to three seconds after an intertrial interval (ITI) of six seconds. A peck to the initialization stimulus resulted in the presentation of one of the eight sample stimuli for a fixed duration of one second on the center key. Following stimulus presentation, the trial continued either with or without a delay period. In the task without a delay, the animals could immediately proceed to the next experimental phase when the confirmation key was pecked. In the delayed condition, a working memory interval of three seconds followed the stimulus during which the placeholder stimulus was presented. The different conditions were tested blockwise so that the sessions were either "non-delayed" or "delayed" sessions. Since the animals were first trained in the "non-delayed" condition, this

part of the experiment was not counterbalanced. After the three second working memory interval, a peck on the placeholder stimulus cleared the center key and activated the choice keys left and right to the placeholder. Depending on the stimulus identity, a choice A or B had to be executed within a time interval of three seconds. The association of the stimuli with choice A and B was counterbalanced across all experiments. Thus, while all animals were exposed to same stimulus set, some animals had to for example make a left choice for a specific stimulus while other animals had to make a right choice for the same stimulus. Depending on the experiment, this choice was either a spatially fixed choice to the left or to the right (experiment 1, 3 and the electrophysiological experiment) or a blue or yellow target color that randomly appeared on the left or right response key (experiment 2, see Fig. 1b). In case of a correct choice, the animals were rewarded by having access to food for two seconds. The ambient light in the operant chamber was turned off for two seconds when a wrong decision was made.

In the behavioral experiments, effects of stimulus-specific spatial and color information on working memory performance were systematically investigated (all experimental procedures are given in Fig. 1b). In experiment 1, we were interested in differences in task performance due to stimulus-specific visual information under uniform reward contingencies with 100% reward probability for all stimuli using either a simple paired association task (non-delayed condition) or an additional working memory component (delayed condition). Experiment 2 replicated the experimental procedure of experiment 1, but additionally dissociated the upcoming choice from a specific spatial location to account for potential spatial memory effects. In experiment 3, we aimed to control for the potential effects of altered reward contingencies on working memory performance. In this experiment, two complex stimuli with opposing choice contingencies received a reward in 30% of all correct choices. The other two complex stimuli were associated with a reward probability of 90%. Simple stimuli received an intermediate amount of reward, i.e. a 50% reward probability in this experiment.

**Behavioral data analysis**. To enter further behavioral analysis, all sessions had to meet the following criteria: At least 70% correct choices had to be reached to ensure that choices reflected learned behavior. Behavioral performance of the animals was analyzed using a linear mixed model with "delay" (yes/no as levels) and "stimulus" (levels: complex stimuli, simple stimuli) as fixed effects and the individual animal as a random effect for every condition. This analysis was computed to identify how the task performance changes over time in relation to the stimulus-specific information. All post hoc corrections were calculated using Bonferroni's method. All tests were two-sided. We also quantified effect sizes for post hoc comparisons (Cohen's $d$). Behavioral data originating from electrophysiological recording sessions were analyzed separately and behavioral results obtained in the electrophysiological experiments were independent of all other behavioral tests.

**Surgery**. After reaching stable behavioral performance for both stimulus classes, microdrives for electrophysiological recordings were implanted. Pigeons were initially anesthetized with an injection of Ketamine (Ketavet, 100 mg/mL; Zoetis, Germany) and Xylazine (Rompun, 20 mg/mL; Bayer, Germany; ratio 7:3; 0,075 ml/100 g body weight). Feathers overlying the head were removed and subjects were fixated in a stereotactic apparatus. Anesthesia was maintained by constant exposure to Isoflurane (Forene, 100% Isoflurane; Abbot, Germany). Once reflexes were

tested negatively for pain perception, the scalp was cut and retracted. Stainless steel screws (A2 stainless steel 0–80 × ½ inch Phillips Countersunk screws) were placed in the skull to serve as anchors for dental acrylic. Small craniotomies were drilled above the NCL (coordinates AP + 5.5 and ML ± 7.5[63]). After transection of the dura mater, electrode tips were lowered directly below the brain surface. Custom-built implants[64,65] were anchored with dental acrylic (Omniceram évolution flow; Omnident, Germany) and another hole was drilled in which a ground electrode was inserted (Teflon-coated silver wire, Ø = 75 µm, Science Products, Hofheim, Germany). The incised skin was covered with antibiotic balm (Fucidine, 20 mg/g Natriumfusidat; Leo Pharma A/S, Danmark) and sutured. Following surgery, all animals received analgesic injections with Carprofen (Rimadyl, 50 ml/mL Carprofen; Zoetis, Germany) and antibiotic powder treatment (Tyrosur, 1 mg/g Tytothricin; Engelhard Arzneimittel, Germany) for three days and were allowed to recover for at least ten days.

**Electrophysiology**. Recordings of neural signals and their analysis followed the procedures described in detail in Starosta et al.[66]. In brief, neural signals were recorded by fifteen 40 µm formvar-insulated nichrome wires (impedances < 0.01 MΩ; California Fine Wire, Grover Beach, USA). One 75 µm nichrome wire (Franco Corradi, Milan, Italy) was used as a reference. All electrodes were connected using microplugs (Ginder Scientific, Nepean, CA, USA). The electrodes were advanced by turning the drive screw at least a quarter revolution (~60 µm) 15 minutes before each recording session. Signals were amplified (400×), band-pass-filtered (0.5 to 5 kHz) and, digitized using an analog-to-digital converter (sampled at a frequency of 22 kHz; Alpha Omega, Nazareth Illit, Israel). Data were recorded using Alpha Lab SnR recording software (Alpha Omega, Nazareth Illit, Israel) and analyzed using Spike2 (Version 7.06; Cambridge Electronic Design, Cambridge, UK). Suspected neural spikes were identified through amplitude thresholds, sorted manually using principal component analysis (PCA) and cluster correlations (Spike2, Version 7.06; Cambridge Electronic Design, Cambridge, UK). The classification as single units was performed using a custom-written MATLAB code and required a clearly dissociated distribution in the PCA space, absence of brief interspike intervals (<4 ms) and, symmetrical distributions of spikes with minimal and maximal peak amplitudes (MLIB toolbox, Maik Stüttgen, MATLAB central file exchange # 37339). Signal-to-noise-ratios (SNR) of at least 2.0 were mandatory (SNR was computed by dividing the peak-to-peak amplitude against the noise band distribution ranging from the 2.5th and 97.5th percentile of the noise band). Additionally, we manually controlled the channels for spike events during pecking movements and computed the number of spikes occurring in an interval of ± 20 ms surrounding a key peck in a peri-peck time histogram (PPTH). Units that specifically fired during the temporal interval surrounding key pecks were excluded from further analysis to avoid potential pecking-artifacts[67].

**Neural data analysis**. Raw spike counts were filtered with an exponentially modified Gaussian kernel with 100 ms as standard deviation and a 100 ms time constant of the exponential function for the computation of spike density functions (SDF). The onset of the sample stimulus served as the temporal origin ($t = 0$) for the sample phase. Relevant pre- and post-timings were calculated individually depending on the duration and nature of the phase. For the overall data analysis, only neural activity during correct trials was analyzed. We also computed activity in error trials however to identify whether neural activity was behaviorally relevant.

**Task related activity—cell classification.** To investigate task-related activity, we computed two-way ANOVAs with factors "value" (two levels, "high" and "low") and "choice" (left or right) separately for the complex and simple stimuli. The stimulus classes were analyzed separately as we identified major performance differences between them in the preceding behavioral experiments. Value was defined in terms of reward probability in the case of the complex stimuli, i.e. high vs. low reward probability. For the simple stimuli, value was defined through the distinct discrimination difficulty at a constant reward probability. Low contrast stimuli led to lower behavioral performances and consequently yielded lower net outcomes, i.e. fewer reward events across the session compared to the high contrast stimuli.

The two-way ANOVA was applied as a sliding window analysis with 250 ms window and 100 ms step sizes from 500 ms before the sample onset until the end of the delay phase. The outcome phase was not further investigated as reward modulation has been described in the NCL in many different aspects before, such as reward prediction error[66,68,69], reward amount, and subjective reward value[70,71]. We quantified the information carried by neurons using a measure of effect size ($\eta_p^2$, reflecting the percent of explained variance), derived from our ANOVA using the following formula:

$$\eta_p^2 = \frac{SS_{Effect}}{SS_{Effect} + SS_{Error}} \qquad (9)$$

Here, $SS_{Effect}$ reflects the sum of squares of the observed effect and $SS_{Error}$ reflects the sum of squared differences between each observation and the group mean. In general, effect sizes are regarded as "small" for values of $0.02 < \eta_p^2 < 0.1$, moderate for values of $0.1 < \eta_p^2 < 0.2$ and "large" for values of $\eta_p^2 > 0.2$ see ref. [72]. To solely account for relevant modulation and rule out accidental fluctuations in spike rates, we chose a threshold of $\eta_p^2 \geq 0.03$ that occurred at least in two consecutive analysis windows. This can be regarded as a conservative measure as every above-threshold time window offered highly significant results in the ANOVA ($p < 0.01$).

Using the effect sizes of the ANOVA, we categorized NCL neurons into non-mutually exclusive categories:

1. If a neuron exhibited above threshold effects for the factor choice, it was categorized as **choice-related activity**.
2. If a neuron exhibited above threshold effects for the factor value, it was categorized as **value-related activity**.
3. If value, choice and interaction effects occurred at the same time, a single neuron represented one single stimulus in particular at a certain point in time during the task. If a neuron displayed this effect size pattern, it was categorized as **stimulus-related activity**. This classification of stimulus-related activity was previously used in Rainer et al.[35].
4. If a neuron exhibited above threshold effects for the interaction between value and response, it was categorized as displaying **interaction activity**.

To allow for a comparison between the sample and delay phase and to provide more insight into the temporal dynamics of activity changes, we analyzed 1 s epochs during the trials comprising the sample phase as well as the first, second and third second of the delay.

**Chance-based quantity of active cells per category.** After each neuron's activity pattern had been classified for each experimental phase, we aimed to identify whether the occurrence of a particular pattern was significantly different from chance. To quantify the chance-level estimation of the category analysis, we conducted 200 permutations of the data in which we randomly assigned the groups

to the observed firing rates during the experiment. Here, raw firing rates were no longer associated with for example left and right choices or with high and low reward outcomes as they occurred within the experiments. Instead, the respective group assignments in the ANOVA were randomly shuffled. Thus, only stochastic above threshold effects would be detected in this permutation test. As before, we then categorized the results from each permutation accordingly into the four activity categories (choice, value, stimulus, interaction) during each iteration. Since the permutation data barely ever demonstrated effects above the conservative threshold of $\eta_p^2 \geq 0.03$, we were more liberal in the classification into the stimulus category which usually required the factors value, response and their interaction to display above threshold effects. In the permutation test, only two of the three effects of the ANOVA had to exhibit above threshold effects to be categorized into stimulus-related activity. We then tested the frequency of choice, value, stimulus, and interaction cells in the data for significance by comparing the occurrence of cells in each category in the recorded and the shuffled data using Fisher's exact test. Since the averaged data thus provided a fractional number, we rounded these numbers up to the nearest positive integer to provide a valid input for Fisher's exact test.

**Class comparison using a linear mixed model.** In a further step, we also analyzed qualitative differences between the complex and simple stimuli on the electrophysiological level. To this end, we computed the mean effect size ($\eta_p^2$) for each category per individual neuron during the sample phase and each second of the delay in the trial. The mean effect was calculated through dividing the observed effect of each neuron by the number of recorded neurons (104). This average was always calculated over 10 bins as each individual analysis phase consisted of 10 individual bins per 100 ms duration. The mean effect size observed during the presentation of complex and simple stimuli was then compared for the sample phase and each second of the delay phase using a linear mixed model using the stimulus class as fixed effect and the individual animal and recording session as random effects to account for the hierarchical nature and dependencies of the data. We also quantified effect sizes for post hoc comparisons and estimated the effect sizes between complex and simple stimuli (Cohen's $d$).

**Neural activation schemes.** Finally, we aimed to analyze neuronal multiplexing in the NCL population as the previous analyses were not informative whether activity patterns dynamically changed in single neurons across the trial. To this end, we generated four post hoc categories reflecting the most apparent activity patterns that emerged within the population. In contrast to the previous categorization, categories were mutually exclusive in this analysis.

1. Neurons were categorized as "**early choice**" if they exhibited choice-related activity patterns already during the sample phase and demonstrated neither stimulus, value, nor interaction activity.
2. Neurons were categorized as "**late choice**" if they exhibited choice-related activity patterns only after delay onset and demonstrated neither stimulus, value, nor interaction activity.
3. Neurons were categorized as "**stimulus - no choice**" if they exhibited stimulus, value, or interaction activity, but no choice-related activity patterns during the sample and delay phase.
4. Neurons were categorized as "**stimulus - choice**" if they exhibited stimulus, value, or interaction activity and then dynamically switched to choice-related activity patterns later during the trial.

We then compared the number of neurons that were categorized in the four categories for the complex and simple stimuli using Pearson's $\chi^2$ test. We also tested if observed choice-related effects were subject to change over time during the delay phase (trace decay). To this end, we calculated the average effect for all neurons involved in choice coding (early, late and stimulus-choice neurons) for the complex and simple stimuli in the first and the last second of the delay phase. This averaged effect was then compared using a paired $t$-test.

**Reporting summary**. Further information on research design is available in the Nature Portfolio Reporting Summary linked to this article.

## Data availability

Further information and requests for data, resources, and reagents should be directed to and will be fulfilled by the lead contact Roland Pusch (roland.pusch@rub.de). Numerical source data for figures and graphs in the manuscript can be found in the supplementary data.

## Code availability

Further information and requests for code should be directed to and will be fulfilled by the lead contact Roland Pusch (roland.pusch@rub.de). Our experimental hardware was controlled by a custom written MATLAB code (2022b, 2018a, The Mathworks, Natick, MA, USA) based on the Biopsychology-Toolbox, a free, open-source Matlab-toolbox for the control of behavioral experiments[58]. Data collection was performed using Alpha Lab SnR (Alpha Omega, Nazareth Illit, Israel). Spike sorting was performed using Spike2 (Version 7.06; Cambridge Electronic Design, Cambridge, UK). For characterizing single units we used the free, open-source MLIB toolbox (Maik Stüttgen, MATLAB central file exchange # 37339).

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

## Acknowledgements

This study was funded by the Deutsche Forschungsgemeinschaft (DFG, German Research Foundation) – Projektnummer 316803389 SFB 1280 (projects A01, A14, A19, F01), Gu 227/16-1 and the Research Training Group "Situated Cognition" (GRK 2185/1). J.P. was funded by the German National Academy of Sciences Leopoldina (LPDS 2021-05). We thank Dr. Noemi Rook for her support in the histological analyses.

## Author contributions

R.P. conceived and supervised the experiments, analyzed the data, discussed the result patterns, and wrote the manuscript. J.P. performed the experiments, analyzed the data, discussed the result patterns, and wrote the manuscript. A.H.A. and S.C. performed the stimulus analyses and reviewed the manuscript. J.R. consulted on data analyses, discussed the result patterns, and reviewed the manuscript. C.S.S. conducted behavioral experiments. M.C.S. conceived the experiment, consulted on data analyses, and reviewed the manuscript. O.G. conceived the experiments, consulted on data analyses, discussed the result patterns, and reviewed the manuscript.

## Funding

## Competing interests

The authors declare no competing interests.
