## [Peer review file · Communications Biology]

Reviewers' comments:

Reviewer #1 (Remarks to the Author):

This manuscript reports a single experiment exploring the neural representation of simple and complex visual stimuli involving a working memory task in pigeons. This lab is at the forefront of behavioral neuroscience in the pigeon and the high quality of the research and the writeup reflects this expertise. I recommend publication, but have some questions that should be addressed either in a revision and/or in a cover letter submitted with the revised manuscript.

My major question is why was only a 3-s delay tested in this procedure? Clearly, a 3-s delay is sufficient to necessitate engagement of working memory of the sample prior to the choice phase on a trial, but isn't it possible that the nature of the stimulus (visual simple versus visually complex) may differentially tax working memory and thus interact with delay? Other research in pigeons and other animals has shown an interaction between delay and working memory processes, such as proactive and retroactive interference (Wright et al., 1985 – see reference below). Similar interaction effects could be at play in the current task, such that, for example, a longer delay might result in even the simple visual stimulus showing similar neural components as the complex stimulus as picked up by the neural recordings. Such nuance was left uninvestigated in the current behavioral procedure.

The analyses of the behavioral data failed to report pair-wise comparisons between non-delayed and delayed conditions in many places. It would be interesting to know where, for example, performance on the Simple Delayed condition was significantly worse than on the Simple Non-Delayed condition. A proper 2x2 analysis given the two factors that were manipulated orthogonal to each other (Delay and Stimulus Complexity) should be reported as well, so that an interaction between factors can be assessed.

It was unclear how many behavioral sessions each bird contributed. For example, when we are told that 104 behavioral sessions from 17 animals were analyzed for Experiment 3 (page 10 line 201), is this 104 sessions per animal or 104 sessions after combining from across all animals? It would be nice to know how many sessions on average each animal contributed to each experiment.

Why were no control brain regions which would not be expected to be involved in working memory performance included? Although the difference in neural signatures for simple versus complex stimuli seems to allow for a within-subject control, given the possibility for an interaction between delay length and working memory impact on neural recordings (see my first comment), it would have been nice to show no effect of stimulus complexity or delay in a different brain region that would not be expected to be impacted by these behavioral manipulations.

It was unclear what "Complex pair 1/2" meant in supplementary figures 1-3.

Page 46, line 878, "The different conditions were tested blockwise...". What orders were used and were they counterbalanced? What about acquisition performance? Did pigeons acquire one type of stimulus (simple vs. complex) faster than the other?

Page 47, line 882, what factors were counterbalanced?

Wright, A. A., Santiago, H. C., Sands, S. F., Kendrick, D. F., & Cook, R. G. (1985). Memory processing of serial lists by pigeons, monkeys, and people. *Science*, 229(4710), 287-289.

Reviewer #2 (Remarks to the Author):

This is an interesting and well-performed study that investigates the impact of the complexity of a visual stimulus on the working memory capacity in pigeons. In three behavioural experiments, pigeons showed consistently better working memory performances when stimuli with higher

complexity were used. The first two experiments counterbalanced the positions of the reward on the left and right sides. In these two experiments, all correct trials were rewarded. On the contrary, in the third behavioural experiment, the rewards for correct choices were provided with different probabilities. Finally, in the last experiment, single units recordings were performed from the caudolateral nidopallium (NCL) in three pigeons during the variable rewards task from Experiment 3. As predicted, complex stimuli were maintained better in the working memory than simple stimuli. Only for the complex stimuli the neural signals represented the upcoming choice during the delay period. Since the same neurons represented different parameters, the authors concluded that they were processing multiple signals simultaneously and hence were multiplexing. On the contrary, simple stimuli did not cause multiplexing, and the choices were detectable in their signals only when pigeons could still see the stimuli.

As far as I can see, this is the first time that single-unit representations of complex and simple stimuli at the level of working memory were described, at least for birds. Therefore, I believe the paper will be interesting for the broad readers of "Communications Biology".

The article is well-written and easy to read. The introduction provides enough background information. The illustrations are clear and informative. The discussion is based on the results. I have, therefore, no major issues and only a few minor comments which can be easily addressed.

Minor comments:

- Line 28 in the abstract: please add that your study used pigeons.
- Line 57: Please add which species were used in the referred study. Please be careful not to overgeneralise across different species, as the similarity of the mechanisms is not trivial.
- Lines 225-226: Please replace 'avian prefrontal analogue' with caudolateral nidopallium (or NCL)
- Lines 228-232: The sentence starting with 'Specifically...' can be deleted since it's a repetition from the introduction.
- Discussion: Is multiplexing done by single neurons? In studies using extracellular recordings, even the best isolated single units can still represent a population of neurons which work as a neural network and respond synchronously. To my understanding, only intracellular recordings would overcome this limitation. I would like to see a mention of this potential limitation in the discussion and a little more introduction to and clarification of the concept of neural multiplexing (e.g. in the paragraph at lines 518-530).
- SI Figure 4: please add a photo of a histological section; otherwise, it's not a reconstruction but an illustration of targeted coordinates.

Reviewers' comments:

Reviewer #1 (Remarks to the Author):

Point 1: This manuscript reports a single experiment exploring the neural representation of simple and complex visual stimuli involving a working memory task in pigeons. This lab is at the forefront of behavioral neuroscience in the pigeon and the high quality of the research and the writeup reflects this expertise. I recommend publication, but have some questions that should be addressed either in a revision and/or in a cover letter submitted with the revised manuscript.

Response: We thank the reviewer for their positive evaluation and the recommendation for publication of our study.

Point 2: My major question is why was only a 3-s delay tested in this procedure? Clearly, a 3-s delay is sufficient to necessitate engagement of working memory of the sample prior to the choice phase on a trial, but isn't it possible that the nature of the stimulus (visual simple versus visually complex) may differentially tax working memory and thus interact with delay? Other research in pigeons and other animals has shown an interaction between delay and working memory processes, such as proactive and retroactive interference (Write et al., 1985 – see reference below). Similar interaction effects could be at play in the current task, such that, for example, a longer delay might result in even the simple visual stimulus showing similar neural components as the complex stimulus as picked up by the neural recordings. Such nuance was left uninvestigated in the current behavioral procedure.

Response: The rationale of testing the 3s delay period in our procedure is twofold. On the one hand, as the reviewer correctly pointed out, it is sufficient to necessitate engagement of working memory of the sample prior to the choice phase. On the other hand, it allows to gain the number of trials within a session needed for a robust statistical analysis. Further increasing the delay has detrimental effects on the overall performance of the animals and they repeatedly stop responding in the middle of the behavioral session. In addition, the brain volume that can be sampled in our neuronal recordings is unfortunately limited and the statistical power will decrease when testing additional delay increments. Thus, we had to opt for a timespan for one delay that satisfies both abovementioned aspects. The chosen timespan for the delay in our study is frequently used in other experiments related to avian working memory (e.g. Anderson et al., 2020; Rinnert et al., 2019; Rose et al., 2005).

Anderson, C., Johnston, M., Marrs, E. J., Porter, B., & Colombo, M. (2020). Delay activity in the Wulst of pigeons (*Columba livia*) represents correlates of both sample and reward information. *Neurobiology of learning and memory*, 171, 107214. <https://doi.org/10.1016/j.nlm.2020.107214>

Rinnert, P., Kirschhock, M. E., & Nieder, A. (2019). Neuronal Correlates of Spatial Working Memory in the Endbrain of Crows. *Current Biology : CB*, 29(16), 2616–2624.e4. doi: 10.1016/j.cub.2019.06.060

Rose, J., Colombo, M. (2005) Neural correlates of executive control in the avian brain. *PLoS Biol.* 3: e190. doi: 10.1371/journal.pbio.0030190

We thank the reviewer for pointing out that interaction effects (proactive and retroactive interference) might interact with the delay for the different stimuli used. Indeed, an experiment testing of for example the serial list function for the different stimulus classes we used would be an interesting endeavor and differences between the stimuli could be suggested. In our case, only a single stimulus was employed in each trial. Any interfering effect, as shown in the mentioned study could hence not

be investigated systematically. To highlight the importance of these effects, we included this issue in our discussion. It reads:

“To systematically investigate interfering effects of the different stimulus classes, an experiment as performed by Wright et al. (1985) would be needed. Testing the serial position function of the different stimulus classes under diverse delay conditions would shed further light on differential encoding and the resulting interference pattern for each stimulus class. Additionally, increasing the number of items would inform about the working memory capacities for stimuli of different visual quality.”

Point 3: The analyses of the behavioral data failed to report pair-wise comparisons between non-delayed and delayed conditions in many places. It would be interesting to know where, for example, performance on the Simple Delayed condition was significantly worse than on the Simple Non-Delayed condition. A proper 2x2 analysis given the two factors that were manipulated orthogonal to each other (Delay and Stimulus Complexity) should be reported as well, so that an interaction between factors can be assessed.

Response: The reviewer is correct that the interaction should be resolved for both factors. We now added the results for each behavioral experiment. They read:

“Resolving the interaction for the factor “delay” showed a significant reduction for both complex ($t = 2.62$, $p = 0.001$, $d = 0.61$, 95% CI = [0.15 – 1.08]) and simple stimuli performance ($t = 9.80$, $p < 0.001$, $d = 2.29$, 95% CI = [1.77 – 2.81]).”

“Identically to experiment 1, we again resolved the interaction for the factor “delay”. We found no significant reduction in performance for complex ($t = 1.89$, $p = 0.062$, $d = 0.63$, 95% CI = [-0.04 – 1.31]) but a significant reduction for simple stimuli ($t = 7.46$, $p < 0.001$, $d = 2.50$, 95% CI = [1.71 – 3.82]).”

“As in experiment 2, resolving the interaction for the factor “delay” showed no significant reduction for complex ($t = 0.39$, $p = 0.700$, $d = 0.11$, 95% CI = [-0.67 – 0.46]) but a significant reduction in performance for simple stimuli ($t = 4.99$, $p < 0.001$, $d = 1.41$, 95% CI = [0.82 – 1.99]).”

Point 4: It was unclear how many behavioral sessions each bird contributed. For example, when we are told that 104 behavioral sessions from 17 animals were analyzed for Experiment 3 (page 10 line 201), is this 104 sessions per animal or 104 sessions after combining from across all animals? It would be nice to know how many sessions on average each animal contributed to each experiment.

Response: We thank the reviewer for pointing this out. We now added a supplementary table (SI Table 1) that lists how many sessions came from each animal for each experiment. In addition, we added how many sessions on average each animal contributed to each experiment.

Point 5: Why were no control brain regions which would not be expected to be involved in working memory performance included? Although the difference in neural signatures for simple versus complex stimuli seems to allow for a within-subject control, given the possibility for an interaction between delay length and working memory impact on neural recordings (see my first comment), it would have been nice to show no effect of stimulus complexity or delay in a different brain region that would not be expected to be impacted by these behavioral manipulations.

Response: We thank the reviewer for raising this important point. Indeed, it would have been interesting to have a control brain structure for the recordings in which no such working memory related activity would be expected to illustrate the specificity of the present results for the NCL. A potential structure could for example have been the avian hippocampus and the parahippocampal structures for which no causal links in delayed-match-to-sample paradigms have been found in the past (Colombo et al., 1997). While we agree that such recordings could be insightful, they are unfortunately

not covered by ethical approval as invasive techniques according to German law are not possible if one does not expect any meaningful task contribution in that structure in the first place. We are not aware of a study that recorded from structures in the avian brain without anticipating involvement of that brain region. Many studies from different labs exclusively recorded from the NCL indicating that this is insightful in its own right (e.g., Nieder et al., 2020; Packheiser et al., 2021; Kobylkov et al., 2022; Hahn et al., 2021). As the reviewer points out, the within-subject control nature of the task is in our opinion sufficient to draw inferences. We added a section to the discussion to point out that investigating other brain structures within the same experimental paradigm will be important to draw more specific conclusions about the NCL. It reads:

“While our study provides interesting insights how stimulus complexity affects working memory performance on the behavioral level and about the neural correlates associated with this phenomenon, our study is subject to a few limitations that should be acknowledged. On the one hand, we did not record from neural structures other than the NCL in the task and thus our study does not inform how specific the observed neural responses are to the NCL. Future studies should be conducted in the same behavioral framework while recording from other (in the best case multiple) brain structures to identify if the differentiation between complex and simple stimuli is selectively represented in the NCL.”

Hahn, L. A., Balakhonov, D., Fongaro, E., Nieder, A., & Rose, J. (2021). Working memory capacity of crows and monkeys arises from similar neuronal computations. *Elife*, 10, e72783.

Kobylkov, D., Mayer, U., Zanon, M., & Vallortigara, G. (2022). Number neurons in the nidopallium of young domestic chicks. *Proceedings of the National Academy of Sciences*, 119(32), e2201039119.

Nieder, A., Wagener, L., & Rinnert, P. (2020). A neural correlate of sensory consciousness in a corvid bird. *Science*, 369(6511), 1626-1629.

Packheiser, J., Donoso, J. R., Cheng, S., Güntürkün, O., & Pusch, R. (2021). Trial-by-trial dynamics of reward prediction error-associated signals during extinction learning and renewal. *Progress in neurobiology*, 197, 101901.

Point 6: It was unclear what “Complex pair ½” meant in supplementary figures 1-3.

Response: We added information and changed the figure labels as this was indeed ambiguous. In our study, we used two pairs of complex stimuli, each comprising of one stimulus requiring a left and one stimulus requiring a right choice. In the main text, we averaged across both stimulus pairs but showed the results for each pair separately for maximal transparency. We added this information to the figure captions and changed the figure labels to provide more clarity.

Point 7: Page 46, line 878, “The different conditions were tested blockwise...”. What orders were used and were they counterbalanced? What about acquisition performance? Did pigeons acquire one type of stimulus (simple vs. complex) faster than the other?

Response: Due to the training protocol used in this study, the blocks were not counterbalanced as we always trained the animals without a delay until they reached the performance level criteria outlined in the manuscript. This information was added to the manuscript. We however do not believe that the non-balancing of “non-delayed” and “delayed” blocks affected our results as for example pigeons that were used for recordings were trained on the three second delay prior to implantation of the electrodes and were then again trained using a non-delayed condition after recovery before being subjected to the final experimental paradigm in the recording sessions. Here, the animals performed generally equally well without an imposed delay suggesting that the exposure to the delayed condition does not

impact performance in the non-delayed condition. Regarding learning differences, we did not analyze this data in detail as our study focused on working memory performance. Unfortunately, any data prior to reaching the learning criterion was not systematically kept for further analysis due to data storage limitations. We however believe that this could be an interesting question per se for future studies. However, a detailed analysis of learning effects is beyond the scope of our initial research question.

Point 8: Page 47, line 882, what factors were counterbalanced?

Response: We counterbalanced the left and right (experiment 1, 3 and recordings) or blue and yellow associations for each stimulus across different animals. We added a clarifying sentence to the methods.

It reads:

“Thus, while all animals were exposed to same stimulus set, some animals had to for example make a left choice for a specific stimulus while other animals had to make a right choice for the same stimulus.”

Wright, A. A., Santiago, H. C., Sands, S. F., Kendrick, D. F., & Cook, R. G. (1985). Memory processing of serial lists by pigeons, monkeys, and people. *Science*, 229(4710), 287-289.

Reviewer #2 (Remarks to the Author):

Point 1: This is an interesting and well-performed study that investigates the impact of the complexity of a visual stimulus on the working memory capacity in pigeons. In three behavioural experiments, pigeons showed consistently better working memory performances when stimuli with higher complexity were used. The first two experiments counterbalanced the positions of the reward on the left and right sides. In these two experiments, all correct trials were rewarded. On the contrary, in the third behavioural experiment, the rewards for correct choices were provided with different probabilities. Finally, in the last experiment, single units recordings were performed from the caudolateral nidopallium (NCL) in three pigeons during the variable rewards task from Experiment 3. As predicted, complex stimuli were maintained better in the working memory than simple stimuli. Only for the complex stimuli the neural signals represented the upcoming choice during the delay period. Since the same neurons represented different parameters, the authors concluded that they were processing multiple signals simultaneously and hence were multiplexing. On the contrary, simple stimuli did not cause multiplexing, and the choices were detectable in their signals only when pigeons could still see the stimuli.

As far as I can see, this is the first time that single-unit representations of complex and simple stimuli at the level of working memory were described, at least for birds. Therefore, I believe the paper will be interesting for the broad readers of "Communications Biology".

The article is well-written and easy to read. The introduction provides enough background information. The illustrations are clear and informative. The discussion is based on the results. I have, therefore, no major issues and only a few minor comments which can be easily addressed.

Response: We are very grateful for the reviewer's positive assessment of our study and its suitability for the journal.

Minor comments:

Point 2: - Line 28 in the abstract: please add that your study used pigeons.

Response: We added this information to the abstract.

Point 3: - Line 57: Please add which species were used in the referred study. Please be careful not to overgeneralise across different species, as the similarity of the mechanisms is not trivial.

Response: We agree with the reviewer that this information is important for context of the state-of-the-art. All reported studies were conducted in humans. We therefore added this information prior to reporting the results from these studies. The sentence reads:

“Across a series of behavioral studies in humans, preliminary evidence has been collected that the stimulus material impacts working memory performance.”

Point 4: - Lines 225-226: Please replace ‘avian prefrontal analogue’ with caudolateral nidopallium (or NCL)

Response: We changed the phrasing accordingly.

Point 5: - Lines 228-232: The sentence starting with ‘Specifically...’ can be deleted since it’s a repetition from the introduction.

Response: In accordance with the reviewer’s suggestion, we deleted the sentence.

Point 6: - Discussion: Is multiplexing done by single neurons? In studies using extracellular recordings, even the best isolated single units can still represent a population of neurons which work as a neural network and respond synchronously. To my understanding, only intracellular recordings would overcome this limitation. I would like to see a mention of this potential limitation in the discussion and a little more introduction to and clarification of the concept of neural multiplexing (e.g. in the paragraph at lines 518-530).

Response: The reviewer is correct that there is no absolute certainty irrespective of the sorting procedure unless using intracellular recording techniques. We added this aspect as a limitation to our discussion. We also provided a bit more context on the idea of multiplexing as suggested by the reviewer. The sections read:

“Neural multiplexing refers to a neuron’s capacity to be activated by multiple different task relevant parameters as opposed to highly specialized neurons (Meister et al., 2013) likely due to being embedded in diverse neural ensembles.”

“On the other hand, we cannot rule out that the observed multiplexing was due to insufficient isolation of single units. However, given our strict sorting criteria and the observation of multiplexing in a large number of neurons in the NCL population for complex stimulus trials, we deem it unlikely that this strongly affected our dataset.”

Point 7: - SI Figure 4: please add a photo of a histological section; otherwise, it’s not a reconstruction but an illustration of targeted coordinates.

Response: We added a photo of a histological section to the supplementary figure. And adjusted the figure legend accordingly. It reads:

“SI Figure 4. Histological analysis of electrode tracks for each individual pigeon subjected in the study. A) Sagittal view of the pigeon brain (Güntürkün et al., 2013) including the range of coronal planes of the electrode positions. **B)** Example of the electrode track reconstruction for pigeon # 182 in a Nissl stained brain slice cut at 40µm. **C)** Schematic electrode track reconstruction for all pigeons used in this study collapsed on a coronal section of the pigeon brain. Drawings are based on the pigeon brain atlas by Karten and Hodos (1967). Gray lines indicate the location of the cannula containing the electrodes for each individual pigeon (Pigeon # 182, # 850 and # 666). For all three pigeons, the electrode tracks were located within the borders of the NCL as defined by Herold et al. (2012). A: anterior; AP: anterior-posterior axis; D: dorsal; NCL: nidopallium caudolaterale; P: posterior; V: ventral.”

REVIEWERS' COMMENTS:

Reviewer #1 (Remarks to the Author):

The authors were very thoughtful and responsive to my comments on the initial draft. I am happy to recommend acceptance.